# Spatial Distribution and Habitat Selection of Sarda Cattle in a Silvopastoral Mediterranean Area

**DOI:** 10.3390/ani12091167

**Published:** 2022-05-02

**Authors:** Marco Acciaro, Andrea Bragaglio, Marco Pittarello, Gian Marco Marrosu, Maria Sitzia, Gabriele Sanna, Mauro Decandia, Simonetta Bagella, Giampiero Lombardi

**Affiliations:** 1AGRIS Sardegna, S.S. Sassari-Fertilia 291, Km 18.6, 07100 Sassari, Italy; gmarrosu@agrisricerca.it (G.M.M.); msitzia@agrisricerca.it (M.S.); gabsanna@agrisricerca.it (G.S.); mdecandia@agrisricerca.it (M.D.); 2Dipartimento di Medicina Veterinaria, Università degli Studi di Bari Aldo Moro, sp Casamassima, Km 3, 70010 Valenzano (BA), Italy; andrea.bragaglio@uniba.it; 3Department of Agricultural, Forest and Food Sciences, University of Torino, 10095 Grugliasco, Italy; marco.pittarello@unito.it (M.P.); giampiero.lombardi@unito.it (G.L.); 4Dipartimento Di Chimica E Farmacia, University of Sassari, 07100 Sassari, Italy; bage@uniss.it

**Keywords:** Low-cost GPS collar, livestock residence index, preference index

## Abstract

**Simple Summary:**

Silvopastoral systems support multiple uses, such as cattle grazing, timber harvesting and the provision of many ecosystem services. The management of livestock movement patterns plays a pivotal role in the sustainable use of silvopastoral systems. Uneven livestock distribution can lead to over- and under-grazed areas, negatively affecting plant and animal diversity, as well as ecosystem services. This study was conducted in a Mediterranean silvopastoral area grazed by autochthonous Sarda cattle to determine the spatial distribution and habitat selection of the cows, who were fitted with GPS tracking collars for this purpose. The total time spent by the animals in different areas was mapped to show the spatial distribution of the cattle in the landscape. Moreover, a preference index was computed for different areas and across different seasons. Overall, the areas where the animals drank and received supplementation were strongly preferred, whereas areas with predominantly rocks were strongly avoided. Grasslands were normally used in proportion to their presence in the area. Forest area was frequented by the cows more in the spring and the summer. These results, representing the first findings concerning Sarda cow grazing in silvopastoral areas, could help farmers to implement actions that help exploit the area more evenly by cows, limiting over- and under-grazing.

**Abstract:**

The beef livestock system in Sardinia is based on suckler cows, often belonging to autochthonous breeds, such as the Sarda breed, and they often graze silvopastoral areas. Besides beef meat, silvopastoral systems (SPSs) provide several Ecosystem Services (ESs), such as timber provision, harvested as wood, and watershed protection. Livestock distribution is a critical factor for the sustainable use of SPSs (e.g., to avoid uneven grazing patterns) and information on patterns of spatial use are required. A study was conducted to determine: (i) the spatial distribution and (ii) the habitat selection of Sarda cattle grazing in a Mediterranean silvopastoral area. Over different seasons, 12 free-roaming adult Sarda cows were fitted with Global Positioning System (GPS) Knight tracking collars to calculate an index mapping of the incidence of livestock in the landscape (LRI) and a preference index (PI) for different areas. Since the PI data were not normally distributed, the Aligned Rank Transform (ART) procedure was used for the analysis. LRI was able to represent the spatial variability in resource utilization by livestock as a LRI map. Overall, the areas where the animals drank and received supplementation were strongly preferred by the cows, reaching PI values in the summer of 19.3 ± 4.9 (median ± interquartile range), whereas areas with predominantly rocks were strongly avoided (the worst PI value in the spring was 0.2 ± 0.6). Grasslands were, in general, used in proportion to their presence in the area, with slightly increased use in the spring (PI 1.1 ± 0.5). Forest area was avoided by cows, except in the spring when it was used in proportion to their presence in the area.

## 1. Introduction

The beef livestock system in Sardinia (Italy) is similar to others in the Mediterranean area [1,2,3]: the cows mainly belong to autochthonous breeds, characterized by an excellent ability to exploit natural forage resources, coupled with a good maternal ability; they normally graze the mountainous and hilly areas, often with the presence of trees (silvopastoral areas) [4,5,6,7,8,9]. The calves are weaned at about 6–7 months old and are then sold to the fattening centers. In a pasture-based system such as this, silvopastoral systems (SPSs) are commonly identified as a source of ecosystem services (ESs), including the provision of beef meat, wildlife habitat and biodiversity and watershed protection, among others [10,11,12,13]. Also, dairy systems (specifically conventional or organic types), even though they are partially pasture-based, provide several ecosystem services [14]. Cattle are often released and allowed to roam freely. Free-roaming livestock have resulted in more selective and spatially heterogeneous grazing distributions [15]; this uneven livestock distribution has resulted in widespread over- and under-grazing. As stated by Bailey [16], the management of movement patterns in livestock is critical for SPSs’ sustainable use. Ganskopp and Bohnert [17] point out several attributes affecting livestock distribution, including the species or breed, the need for escape or concealment, the reproductive status, the proximity of drinking water, the density of woody vegetation, the plant community composition and its associated effects on forage quantity and quality, diurnal temperature dynamics and prevailing winds [18]. Hessle et al. [19] identified the season as an important factor in the interactions between herbivores and their foraging behavior. Season is also related to the length of daylight, which, according to the antipredator theory [20], impacts foraging behavior. Nowadays, numerous tools are being employed to assist with the manual observation of grazing behavior [21]. Global Positioning System (GPS) technology allows researchers to continuously track cattle locations, obtaining large amounts of data over short sampling intervals and large spatial scales [22]. The affordability of modern GPS tracking collars allows for accurate and consistent measurement of the distribution of livestock in the landscape, which helps evaluate different aspects of grazing management [23]. The objectives of this study were:(i)to detect the spatial distribution (Livestock Residence Index) and the selection of the vegetation communities (expressed through a preference index (PI)) of Sarda cows, fitted with Global Positioning System (GPS) tracking collars and grazing a Mediterranean silvopastoral area.(ii)to estimate the spatial distribution (Livestock Residence Index) and the selection of the vegetation communities (PI) of Sarda cattle fitted with GPS tracking collars over the different seasons.

## 2. Materials and Methods

### 2.1. Experimental Site 

This study was conducted in the experimental farm of the Agricultural Research Agency of Sardinia (AGRIS Sardegna, Oristano, Italy), located in Monte Sant’Antonio (40°14′10′’ N, 8°42′31′’ E, Macomer, Italy) at 690 m a.s.l. The study was carried out respecting the rules, principles and specific animal care and welfare guidelines of Italian law (Gazzetta Ufficiale, DL no. 116, 27 January 1992). The fenced 54 ha study area (Figure 1) is gentle terrain, characterized by an average elevation of 695 m (range: 730–660 m a.s.l.) and an average slope of 10.7% (range: 0.5–36.3%). 

The climate is Mediterranean, characterized by: average maximum temperature Tmax = 29.6 °C; average minimum temperature Tmin = 1.9 °C; total annual rainfall = 688 mm (http://www.sar.sardegna.it/, accessed on 25 January 2022). No extreme temperatures or precipitation events occurred for the duration of this trial (Table 1). 

To better represent the spatial distribution of Sarda cows in the experimental area, we identified (Figure 2):-three vegetation units (*sensu stricto*), which were easily distinguishable on a physiognomic-structural basis: (I) deciduous oak woods dominated by *Quercus pubescens* s.l.; (II) shrub-encroached grasslands dominated by *Rubus hulmifolius* and *Pteridium aquilinum* (bracken), with a sporadic presence of trees; (III) grasslands dominated by grasses and other herbaceous species (e.g., forbs, legumes). No assessments of the homogeneity of forage production within units were carried out during the experiment.-rocky areas, where rock covers more than 30% of their surface (IV).-the area around the water point and the feed supplement, consisting of occasionally administered hay; this area was identified as feedwater. The three vegetation units, the rocky areas and the feedwater areas were visually determined according to the plant composition and experience from previous years. There is only one water point within the study area, located in the southern part of the experimental paddock. Although a strong preference for the feedwater area was expected, we also preferred to consider this area in the analyses to verify and quantify its presumed overuse.

As a whole, the experimental area covered 54.3 ha. Deciduous woods represented more than 50% of this area, whereas grasslands were limited to a unique, small patch (Figure 1 and Figure 2, Table 2).

### 2.2. Animals and GPS Data 

The monitoring of animals was conducted during four sampling periods (one per season): from 26/February/19 to 03/March/2019 (winter), from 06/April/19 to 18/April/2019 (spring), from 16/July/19 to 24/July/2019 (summer) and from 26/September/2019 to 16/October/2019 (autumn). The area was grazed by a Sarda cow herd (N = 12, 430 ± 62 kg average live weight ± s.d., 11.6 ± 3.3 average years old ± s.d.), conducted together with the calves from the calving period (October-November 2018) to the weaning period (June–July 2019), with a stocking rate of 128 ± 16 kg live weight ha^−1^ (average live weight ± s.d.). The Sarda breed is a small-to-medium-sized local breed, well-adapted to the harsh environment of Sardinian hilly and mountainous areas. Three cows, randomly selected from the herd each season, were equipped with Knight GPS collars [24] and scheduled to record positions every 3 min. The Knight GPS collar uses the igotU GT- 600^®^ GPS unit and a recharge battery. The accuracy of the igotU GPS logger has been previously reported [25], with a location error < 10 m and a standard deviation of 3.04 m. During the winter sampling period, the animals were supplemented with 5 kg head^−1^day^−1^ of natural pasture hay (Dry Matter DM 86.4%, CP 7.11% DM basis, NDF 68.5% DM basis, ADL 4.5% DM basis) as a result of the low pasture availability, which is typical of this period. Although this supplementation is an element that modifies the experimental design, since winter was the only season in which hay was administered, it was decided not to exclude this season from the analysis, as these practices are widespread in farms characterized by this livestock system. Tracking data recorded during the day of the collar placement and the day of removal were not used in our analyses. The fix rate (given by the ratio between the number of positions recorded and the scheduled number of fixes) was calculated for each collar and each season. Afterwards, raw data points for each season were mapped as a livestock residence index (LRI) [26] using spatial analyst tools in the QGIS program (v. 3.10.14 “A Coruña”). The LRI is an assessment of the total time spent by the animals in a certain area, which is useful to map the incidence of livestock in the landscape. Using the QGIS program, a 50 × 50-m grid was overlayed on the GPS locations and for each season, the number of locations within each grid cell were counted. The LRI for any given grid cell x (LRIx) was calculated using:LRIx (n) = ∑x Raw GPS locations count/∑n∑x Raw point count
where n is the number of grid cells in the entire trial field. The proportion of points within each grid cell was calculated and shown as LRI maps for each season. To evaluate cattle preference [15,27,28], the preference index (PI) was computed daily as the proportion of GPS records within each area (VUs, rocky areas and feedwater areas) divided by the proportional area covered within the total study area. A 95% confidence interval was calculated for each PI value. The lower confidence limit of PI > 1 indicated preferential selection, while values <1 for the upper confidence limit indicated that cows used that zone proportionally less than its availability in the area. If the value of 1 was within the confidence interval of the PI value, cows used a particular zone in proportion to its presence. The PIs over the seasons were calculated. 

### 2.3. Statistical Analysis

The Aligned Rank Transform (ART) procedure was used to compare PIs across different seasons as the data were not normally distributed. Daily PIs were analyzed using ARTool procedure of R software version 3.3.2 (The R Development Core Team, 2016) [29,30], with zone and season as fixed effects. Post hoc pairwise comparisons were also conducted [31], and F tests determined differences between treatments. Tukey’s multiple comparison test was appropriately applied to evaluate pairwise comparisons between treatment group means. Treatment differences with a *p*-value less than or equal to 0.05 were considered significantly different, unless indicated otherwise.

## 3. Results and Discussion 

### 3.1. Livestock Residence Index (LRI)

The cows equipped with GPS during the study accounted for 25% of the whole herd. Considering that the herd was homogeneous in age, sex and needs (thus fostering the social cohesion), stable in its composition (not subject to changes in its components for several years), and according to [26], a small number of collars can be sufficient for “mob-monitoring”, it appears that the number of animals used may be sufficient for the purposes of this work. After selecting the data according to the GPS collar manufacturer’s instructions, bad data were eliminated based on the animals’ speed, distance traveled between points and time between points; a total of 90104 valid animal positions, recorded during the deployment periods, were kept. The GPS fix rate is shown in Table 3. A collar in the spring recorded an unsatisfactory fix value (44.2 ± 16.4%) and therefore, the data were discarded from the analysis. Without the data from this collar, the fix rate value in the spring was 94.0 ± 2.8%. This favorable result may be due to the short, scheduled fix interval that, as stated by the designer of the collars used in this study [24], allowed the logger to stay in contact with satellites, acquiring fixes by preventing lost data due to failure. 

Figure 3 shows the processed GPS locations mapped as LRIs on 50 × 50 m grid cells. As a GPS fix was collected in a high percentage of attempts (Table 3) and the standard deviation (13.4 s) of the fixes was low, each point was assumed to represent equal time portions. The derived LRI map of the trial field (Figure 3) revealed a spatial variation in residence time. The cows appeared to prefer the feedwater area of the paddock (where the water point is located and the feed supplement is supplied) and appeared to avoid the rocky glades area. Moreover, the maps revealed that the experimental area was more evenly frequented in the spring, revealing a broader use of the entire area. In the summer, the tracked cows were shown to spend a lot of their time near the water (feedwater area) and in the northern and northwestern areas of the paddock; the cows were probably seeking shade under the trees in this location, as this was the highest area of the paddock (724 m.a.s.l.) and exhibited cooler temperatures. According to other authors [32,33], this shade-seeking and its relationship to temperature are well-known driving factors in areas utilized by livestock. Furthermore, trees, shrubs and variable terrain in rangeland allow livestock to seek out sites with more favorable climatic conditions. 

### 3.2. Preference Indexes

The preferences of tracked cows (PI) for the different areas over the different seasons are shown in Table 4 and Table 5. In three out of four seasons, the feedwater area was strongly preferred and the rocky areas were avoided (Table 4), confirming the LRI map. Nevertheless, some exceptions have been recorded. The expected and predictably strong preference of cows for the feedwater area was confirmed during periods of forage shortage and low rainfall (i.e., autumn, winter and summer). As stated by Bailey [16], in these seasons, livestock gathered and spent most of their time in areas near water, as endorsed by the highest PI values to the feedwater area (Table 4). An excessive amount of time in these areas could reduce vegetation cover and increase soil erosion, a condition which needs to be closely monitored. On the contrary, in the spring, this area was used in proportion to its presence and more generally, the experimental paddock was more evenly frequented. This is probably linked to the increased presence and greater availability of lush herbage in this season, as is typical in Mediterranean areas [34,35]. During the winter, spring and summer periods, the rocky area was avoided by the cows, as indicated by the upper confidence limit of PI <1 and by its lowest PI values (Table 4); although in the summer, PIs for the rocky areas were not different from shrub-encroached grasslands). The autumn period was an exception, as the rocky area was used in proportion to its presence and characterized by a PI not different than “grasslands” (Table 4). The high percentage of rocky outcroppings in these areas (ca 30%) and the consequent scarce presence of palatable forages probably makes them less sought after by the animals. On the other hand, brushes fruit in autumn, and these are eaten by the Sarda cows when other forage is scarce; this could justify the PI value that was registered. The deciduous oak wood area was used in proportion to its presence in the spring, whereas this area was avoided in the autumn, winter and summer. In these three seasons, the PIs for the deciduous oak wood area and the “grasslands” were not different, confirming that in these periods, the use of the experimental area by the cows was strongly influenced by the feedwater zone (Table 4). The use of deciduous oak wood areas in the spring was probably due to the presence of grazing herbage in the undergrowth. The shrub-encroached grassland areas were avoided in the summer and the autumn, in conjunction with the maximum presence of brackens (Pteridium aquilinum (L.) Kuhn) and usage in proportion to its presence in the other seasons. Bracken is an unpalatable species containing antinutritional factors, such as cyanogen glycosides, antithiamine and compounds with carcinogenic, mutagenic and clastogenic activity [36]. It is known that grazing animals avoid areas far from water and/or dominated by unpalatable species [37]. Generally, the grassland areas were used indifferently by the tracked cows. In the spring, summer and autumn, the animals did not receive supplements, and rainfalls were higher in the spring than in other periods (Table 1); greater rainfall resulted in a greater availability of forage, which is typical of this season in Mediterranean conditions. These conditions resulted in the animals moving more throughout the whole experimental paddock. Furthermore, this makes it possible to compare the cows’ preference for the different areas, without the presence of hay and water points as attractive factors. In this situation, the cows used the different areas (with the exception of the avoided rocky areas) in proportion to their presence and without showing differences in preferences, probably because of the good availability and quality of grass in this season. On the contrary, rocky glades were the only area avoided, and the lower PI confirmed its low attractiveness to cows. 

These results endorse the importance of the water point in the spatial distribution of grazing cows and the lower attractiveness of areas with the presence of brackens.

The results obtained in this work have some practical applications. 

The GPS collars proved to be a cost-effective tool for some research and management questions that involve animal location and distances. Previous work [38] reported data on daily distance travelled, average and maximum daily distance to water of Sarda cows in the same experimental area. The GPS collars might help field managers to assess the area actually used by the livestock, so that the correct stocking rate can be better established. Moreover, better decisions regarding grazing distribution could be adopted by farmers [23,39]. GPS represents an efficient and accurate method of measuring grazing distribution and livestock tracking data can provide managers with information to make more informed decisions [39]. On the other hand, as suggested by [40], the aim of precision livestock management is real-time monitoring to improve livestock productivity and welfare. This aspect, and the need for sophisticated analyses to interpret the data, means the model shown in this work still needs to be refined. 

In agreement with several authors [10,41,42], our results showed the role of the piosphere, identified as the “area near to drinking water”, in driving the behavior patterns of free-ranging herbivores. In the experimental area, hay administration for grazing animals is close to the water point. This practice created an area highly frequented by some animals to the detriment of others. This concentration of livestock, associated with uneven grazing distribution, may cause inconveniences (i.e., increasing soil erosion) and adversely affect wildlife habitats. In the area where we worked, an important percentage of the surface is occupied by bracken (26%, Table 1), which results in a scarce presence of animals; our results also reflected this point. This finding demonstrates the importance of understanding grazing distribution patterns for the calculation of stocking rates, as the area occupied by bracken is infrequently used in practice. On the other hand, the need to improve this part of the pasture is highlighted, even if, in line with some authors [23], it could be preferable that farmers measure actual distribution patterns and adjust stocking rates based on site-specific data. The monitoring of the spatial use of silvopastoral land by cows clearly detected that the areas preferred by livestock (feedwater areas) are overgrazed. Managers have numerous tools to manipulate livestock distribution and resolve concerns of concentrated grazing in specific sites (e.g., water point developments, physical or virtual fencing, strategic supplement placement, herding [43]). In addition, although referring to different environments (i.e., alpine mountain pasture), in some cases the opportunity to implement rotational stocking systems at recommended stocking levels with increased uniformity of grazing and decreased selectivity and consequently less shrub and tree encroachment [15,43] could be evaluated.

## 4. Conclusions 

Based on the use of low-cost GPS collars, the spatial distribution and habitat selection of Sarda suckler cows grazing a silvopastoral Mediterranean area were investigated. The spatial distribution was expressed by index mapping of the incidence of livestock on the landscape (LRI), and habitat selection was calculated through a preference index (PI) for different areas. LRI was able to represent the spatial variability in resource utilization by livestock as an LRI map. Overall, the areas where the animals drank and received supplementation was strongly preferred by the cows, reaching a higher PI value in the summer and the winter, whereas areas with predominantly rocks were strongly avoided. Grasslands were generally used in proportion to their presence, but if non-palatable species were present, they were avoided; for example, this phenomenon occurred in the summer, specifically when the presence of brackens was particularly high. Cows generally frequented woody areas in proportion to their extent. These results, which are the first concerning Sarda cow grazing in silvopastoral areas, allowed us to identify preferred, indifferent and avoided areas, providing information to farmers for the implementation of management operations to counteract prospective, uneven grazing-related damage. 

## Figures and Tables

**Figure 1 animals-12-01167-f001:**
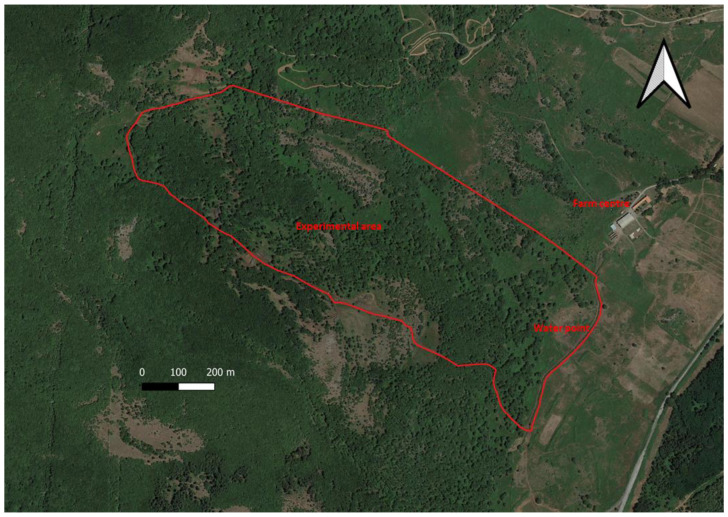
Experimental area.

**Figure 2 animals-12-01167-f002:**
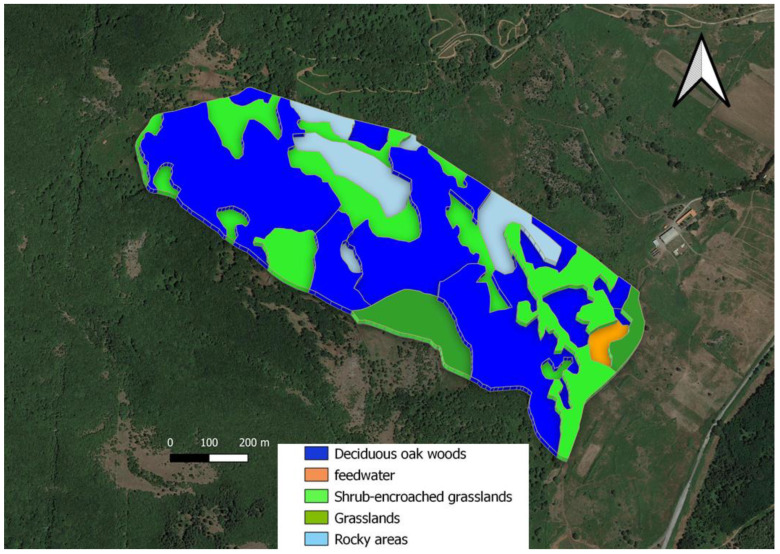
Map of experimental area. I. Deciduous oak woods, dark blue area; II. Shrub-encroached grasslands, bright green area; III. Grasslands, olive green area; IV. Feedwater, orange area; V. Rocky areas, light blue area.

**Figure 3 animals-12-01167-f003:**
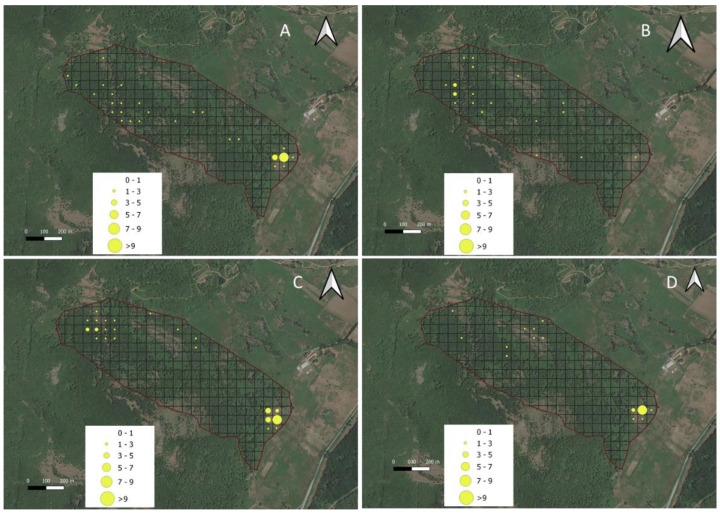
Spatial variation in experimental area used by Sarda cows over the winter (**A**), spring (**B**), summer (**C**) and autumn (**D**) deployment period expressed as a Livestock residence index (LRI X 100) map on a 50 m grid.

**Table 1 animals-12-01167-t001:** Maximum (T max), minimum (T min), average temperature (T avg) (mean ± s.d.) and rainfall patterns (total rainfall, mm) during the experimental periods.

	Experimental Periods	T max	T min	T avg	Rainfall
Winter	26/ February /19 to 03/March/2019	13.6 ± 1.2	2.2 ± 1.6	7.8 ± 0.8	0.2
Spring	06/April/19 to 18/April/2019	16.6 ± 2.5	4.3 ± 1.9	8.5 ± 2.0	52.4
Summer	16/July/19 to 24/July/2019	30.3 ± 2.7	15.6 ± 3.2	22.8 ± 2.7	6.8
Autumn	26/September/2019 to 16/October/2019	22.0 ± 2.5	10.9 ± 1.9	15.9 ± 1.90	14.2

**Table 2 animals-12-01167-t002:** Area and frequency of occurrence (% of the experimental area) of the three vegetation units (VUs), rocky areas and feedwater.

VU	Area of Vegetation Units (Ha)	Fraction of Study Area (%)
Deciduous oak wood	30.3	55.8
Shrub-encroached grasslands	14.1	26.0
Grasslands	3.7	6.9
Rocky area	5.5	10.0
Feedwater	0.7	1.3

**Table 3 animals-12-01167-t003:** Seasonal fix rate value (average ± s.d.) given by the ratio between the number of positions recorded and the scheduled number of fixes (480 positions every 24 h).

	Fix Rate (%)
Winter	94.3 ± 27.6
Spring	79.2 ± 24.8
Summer	89.8 ± 4.0
Autumn	93.1 ± 2.2

**Table 4 animals-12-01167-t004:** Proportion of GPS records and preference index (PI) (with a 95% confidence interval) for three structural physiognomic units of vegetation (VU), rocky areas and feedwater areas in the different seasons of grazing cattle under continuous grazing system in a Mediterranean silvopastoral area (median ± interquartile range).

		n°	Proportion GPS Records	Proportion Area	Preference Index	Lower Limit	Upper Limit
Autumn	VU Deciduous oak wood	21	0.44	0.56	0.78 ± 0.18 b	0.70	0.88
	Feedwater	21	0.16	0.01	11.90 ± 1.50 a	11.4	12.90
	VU Shrub-encroached grasslands	21	0.22	0.26	0.83 ± 0.15 b	0.76	0.91
	VU Grasslands	21	0.06	0.07	0.81 ± 0.38 b	0.66	1.04
	Rocky areas	21	0.10	0.10	1.01 ± 0.89 b	0.67	1.56
Winter	VU Deciduous oak wood	6	0.48	0.56	0.86 ± 0.17 b	0.77	0.94
	Feedwater	6	0.18	0.01	13.80 ± 5.46 a	9.94	15.40
	VU Shrub-encroached grasslands	6	0.28	0.26	1.07 ± 0.44 b	0.85	1.29
	VU Grasslands	6	0.05	0.07	0.79 ± 0.78 b	0.39	1.17
	Rocky areas	6	0.03	0.10	0.26 ± 0.29 c	0.05	0.34
Spring	VU Deciduous oak wood	13	0.55	0.56	0.99 ± 0.22 a	0.87	1.09
	Feedwater	13	0.01	0.01	1.00 ± 5.66 ab	0.00	5.66
	VU Shrub-encroached grasslands	13	0.28	0.26	1.08 ± 0.39 a	0.95	1.35
	VU Grasslands	13	0.06	0.07	0.95 ± 0.99 a	0.65	1.65
	Rocky areas	13	0.02	0.10	0.22 ± 0.49 b	0.01	0.50
Summer	VU Deciduous oak wood	9	0.49	0.56	0.88 ± 0.20 b	0.79	0.99
	Feedwater	9	0.25	0.01	19.30 ± 5.20 a	16.10	21.30
	VU Shrub-encroached grasslands	9	0.17	0.26	0.67 ± 0.43 bc	0.43	0.86
	VU Grasslands	9	0.05	0.07	0.76 ± 0.59 b	0.51	1.10
	Rocky areas	9	0.03	0.10	0.32 ± 0.72 c	0.03	0.76

Different lowercase letters indicate significant differences among areas within the same season (*p* < 0.05).

**Table 5 animals-12-01167-t005:** Effects of seasons on proportion of GPS records and preference index (PI) (with a 95% confidence interval) for three structural physiognomic units of vegetation (VU), rocky areas and feedwater areas of grazing cattle under continuous grazing system in a Mediterranean silvopastoral area (median ± interquartile range).

	Season	n°	Proportion GPS Records	Proportion Area	Preference Index	Lower Limit	Upper Limit
Deciduous oak wood	Autumn	21	0.44	0.56	0.78 ± 0.18 b	0.70	0.87
	Winter	6	0.48	0.56	0.86 ± 0.17 ab	0.94	0.77
	Spring	13	0.55	0.56	0.99 ± 0.22 a	0.87	1.09
	Summer	9	0.49	0.56	0.88 ± 0.20 ab	0.80	0.99
Feedwater	Autumn	21	0.15	0.01	11.90 ± 1.50 b	11.40	12.90
	Winter	6	0.18	0.01	13.80 ± 5.46 b	9.94	15.40
	Spring	13	0.13	0.01	0.99 ± 5.66 c	0.00	5.66
	Summer	9	0.25	0.01	19.30 ± 5.20 a	16.10	21.30
Shrub-encroached grasslands	Autumn	21	0.21	0.26	0.83 ± 0.15 bc	0.76	0.91
	Winter	6	0.28	0.26	1.07 ± 0.44 ab	0.85	1.29
	Spring	13	0.28	0.26	1.08 ± 0.39 a	0.95	1.35
	Summer	9	0.17	0.26	0.67 ± 0.43 c	0.43	0.86
Grasslands	Autumn	21	0.06	0.07	0.81 ± 0.38	0.66	1.04
	Winter	6	0.05	0.07	0.79 ± 0.78	0.39	1.17
	Spring	13	0.06	0.07	0.95 ± 0.99	0.66	1.65
	Summer	9	0.05	0.07	0.76 ± 0.59	0.51	1.10
Rocky areas	Autumn	21	0.10	0.10	1.01 ± 0.89 a	0.67	1.56
	Winter	6	0.03	0.10	0.26 ± 0.29 b	0.05	0.34
	Spring	13	0.02	0.10	0.22 ± 0.50 b	0.01	0.51
	Summer	9	0.03	0.10	0.32 ± 0.69 ab	0.03	0.72

Different lowercase letters indicate significant differences among seasons within the three structural physiognomic units of vegetation (VU), rocky and feedwater area (*p* < 0.05).

## Data Availability

Data available on request.

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
