# Peer review of "Spatial Distribution and Habitat Selection of Sarda Cattle in a Silvopastoral Mediterranean Area"

_animals, 2022, doi:10.3390/ani12091167_

Round 1
Reviewer 1 Report
The main weakness of this manuscript is it overall low clarity. I believe that improve its readibility will help the reader to better understand the main findings and implications. Please avoid long sentences without ponctuations.
Specific comments
L62: Please check the sentence and values. 12-14 month instead of 14-12?
L117: distribution by?
L131: add a comma after whole; the second sentence needs revision... 50% of the experimental?
L142: What do the authors mean by "a mature suckler cow herd"? Please rephrase the whole sentence
L150-155: I would suggest to remove these sentences
L158-159: do you mean a typical feeding practice during this season due to fodder shortages? Please revise
L178-179: What is the meaning of if "each zone was avoided"?
L180: Values of what?
L240: add "s" to period
L242: sentence in bracket ("although...grasslands") is incomplete
L246: make instead of makes
L247:-250: "Two times on the other hand". please rephrase.
L250: what "its" stands for?
L256: Please rephrase "indeed is..."
L261: do not start an affirmative sentence with "is"
L263-267: please rephrase. What this stands for on L265 and L267?
L287-291: sentence too long and not precise. Please rephrase
297: Please rephrase "our results confirm what[43]..."
L303: the following statement is not supported by any data: "In addition, the low profitability of this livestock entreprise..."
Author Response
Please, see the attachement

Reviewer 2 Report
Dear authors, thank you for your manuscript and your work.
There are a few comments I would like you to be adressed
Introduction:
Please try to be more concise, what you would like to report. The introduction should lead into your research questions. e.g. line 56-60, is all this information needed?
Material and Methods:
L117: there are more than three Charatzerizations mentioned in the figure, please adapt the description, as this is not fully clear. Also please adapt the caption of Fig 2. and give more explanation
L157: Was this hay per day or per period?
Number of animals? Where those only n=3 animals per season, which were monitored? Was this always the same animal? This very low number of animals makes the results less significant. For me there is always the question, is this behaviour based on the individual animal, and can this be compared with the other animals? There should be a higher number of animals, to be reported.
Also do those suckler cows have their calves over all 4 periods? Or will they be weaned in between. This will also definitely influence their behaviour. Maybe even more than the seasons....
Figure 3: The colouring is hard to get. Please select different colors, so the markings can be recognized better.
The combination of results and discussion makes the paper difficult to read. please adapt
Reviewer 3 Report
L 90: need to be corrected : inappropriated objective because there is no evaluation of under or overgrazed realised.
L 107: add a scale to the figure 1
L 117 to 128: need to add details on whether or not assessments of the homogeneity of forage production within units are carried out
L 129: add North and scale on figure 2. Colours of Rocky area the colour should be harmonised between the figure and the legend
L 132: there are 2 paches of grassland on the figure rather than an "unique" one
L 180 and 181: to be modified: ...confidence limit, indicated...
L 222: add North and scale on the figure 3
L 259-261: Add details on the presence of "ptaquiloside" and its measurement or remove the sentence
L 284-296: Add details on the measurement or estimation of the forage composition and production of the units under consideration to support the discussion
L 311-313: amend the sentence as no real measurement of overgrazing is carried out to support this claim
L 333: areas in
L 336 - 337 : amend the sentence as no real measurement of grazing damage is carried out to support affirmation of "grazing damage-related"
L 372-373: the reference 4 must be completed
L 377: inappropriate position of the date in the reference 6; need to harmonise the presentation
L 387: the reference 10 must be completed or removed
L 388: inappropriate position of the date in the reference 11; need to harmonise the presentation
L 390: inappropriate position of the date in the reference 12; need to harmonise the presentation
L 423: inappropriate font of the date in the reference 26; need to harmonise the presentation
L 424: inappropriate font of the date in the reference 27; need to harmonise the presentation
L 427: inappropriate font of the date in the reference 28; need to harmonise the presentation
L 430: inappropriate font of the date in the reference 29; need to harmonise the presentation
L 432: reference 30 not cited in the text: to be deleted! Please note that the following reference numbers 31 to 46 must then be corrected in the text and in the list.
L 435: inappropriate font of the date in the reference 31; need to harmonise the presentation
L 437: inappropriate font of the date in the reference 32; need to harmonise the presentation
L 450: inappropriate font of the date in the reference 37; need to harmonise the presentation
L 453: inappropriate font of the date in the reference 38; need to harmonise the presentation
L 459: inappropriate font of the date in the reference 41; need to harmonise the presentation
L 462: inappropriate font of the date in the reference 42; need to harmonise the presentation
L 472: inappropriate font of the date in the reference 46; need to harmonise the presentation
Reviewer 4 Report
Paper is well-written and presents an important topic, although the novelty of the study is low. The factors driving animal behaviour identified by authors have already been extensively described in literature.
Concerns about experimental design:
- the methodology for objectively mapping the different areas of the paddock is not well described.
- some important features, such as slope, are not analysed and should be included at least in the discussion of project results. The same can be said about meteorological conditions of specific days (authors seem to have access to this information, table 1)
- collar missing data are scarce, but it must be stated that this is the situation for all collars, not on average (missing data in a single collar can affect results).
- in line 200 authors talk about valid positions. Which were the criteria to define valid data?
- they monitor 3 animals, which is a large percentage of the small herd, but still very few animals. Authors should discuss about the coherence of the data among animals.
- the LRI was calculated by grid cell, then averaged by type of landscape? Management of this data should be further explained.
- in general, statistical analyses could be sofisticated and discussion may be improved with all the available information about the paddock.
Typo errors:
- line 38: compare PI for different seasons
- line 150-151: Briefly, low-cost GPS data loggers have been used to build
- table 4: letters in column preference index are confusing. Can they be presented in two columns?
- line 333: woody areas in proportion
- line 335: providing information
- line 336-337: uneven grazing-related damage
- references: some are lacking, e.g. 10 and 30. Some are not standarised (complete and abbreviated journal names). Some are repeated, e.g. 42 and 46. This section may be extensively revised.
Round 2
Reviewer 1 Report
Many thanks for your efforts
Reviewer 2 Report
Thank you for following my suggestions and reedit your paper.